# Breaking the Cycle: Perceived Control and Teacher–Student Relationships Shield Adolescents from Bullying Victimization over Time

**DOI:** 10.3390/bs14121198

**Published:** 2024-12-13

**Authors:** Zhongjie Wang, Kaiyuan Lu, Xuezhen Wang, Juanjuan Zheng, Xinyi Gao, Qianqian Fan

**Affiliations:** 1School of Education, Zhengzhou University, Zhengzhou 450001, China; zhongjie1597@zzu.edu.cn (Z.W.);; 2School of Education, Renmin University of China, Beijing 100872, China

**Keywords:** early adolescence, bullying victimization, perceived control, teacher–student relationships, latent growth model

## Abstract

Background: Bullying victimization remains a grave issue in early adolescence. However, existing research often lacks a longitudinal perspective and sufficient attention to protective factors, particularly the dynamic role of teacher–student relationships. Objective: This study explores the longitudinal protective mechanisms against bullying victimization, focusing on the roles of perceived control and teacher–student relationships. Methods: A sample of 1454 adolescents (mean age = 13.63 years, *SD* = 0.76, 51% female) was followed over the course of one year. Latent growth models were employed to examine the developmental trajectories of bullying victimization, perceived control, and teacher–student relationships, and to investigate the longitudinal mediating role of teacher–student relationships in the relationship between perceived control and bullying victimization. Results: Both perceived control and teacher–student relationships showed a consistent increase over time, while bullying victimization declined over time. The intercept of perceived control negatively predicted the intercept of bullying victimization, with this relationship mediated by the intercept of teacher–student relationships. Furthermore, the slope of perceived control affected the slope of bullying victimization solely through the slope of teacher–student relationships. Conclusions: These findings highlight that perceived control serves as a stable protective factor against bullying, while strong teacher–student relationships can further reduce bullying victimization. Enhancing students’ perceived control and fostering supportive teacher–student relationships should be key components of anti-bullying interventions.

## 1. Introduction

Bullying victimization refers to the phenomenon wherein an individual is subjected to prolonged or repetitive mistreatment or harm by one or more peers [1]. Early adolescence is a period of rapid physical and mental growth, making individuals particularly sensitive to social influences [2]. During this period, adolescents heavily rely on social relationships for validation and a sense of belonging, and undergo intense self-exploration [3]. When these needs for positive social connection are not fulfilled, or when the process of self-exploration involves setbacks, early adolescents may become more vulnerable to negative social experiences such as bullying [3]. This developmental stage also coincides with a peak in the prevalence of bullying behaviors, with approximately 30.4% of early adolescents worldwide experiencing bullying [4,5]. Especially, Chinese adolescence coincides with significant academic pressures, particularly the preparation for the Zhongkao (high school entrance exam). The competitive nature of the Chinese educational system, where academic performance is highly valued, often leads to increased social comparison among peers. Adolescents who do not perform as well academically are more likely to feel socially isolated, stigmatized, and may become targets of bullying by their peers. Bullying can lead to a range of adverse outcomes, including increased risk of depression, anxiety, social withdrawal, and even suicidal ideation [6,7]. Given these serious consequences, bullying victimization represents a critical social issue that demands comprehensive intervention and prevention efforts.

Existing research indicates that perceived control and teacher–student relationships may serve as effective protective factors for bullying victimization [8,9]. While longitudinal studies on childhood have explored the role of perceived control in bullying victimization, research on early adolescence remains limited. For instance, children’s perceived control at the age of 5 could significantly predict their experience of bullying victimization at the age of 6 [8]. However, there is a gap in similar research focusing on early adolescents, who differ from children in their social needs, cognitive abilities, and response to external stressors. These developmental differences likely influence how perceived control impacts bullying dynamics. Therefore, it is essential to investigate the role of perceived control in early adolescence, particularly within the context of the increasing academic and social pressures that shape this stage of development.

The longitudinal relationship between teacher–student relationships and bullying victimization remains inconclusive. For example, teacher–student relationships among 13-year-old adolescents could predict their experience of bullying victimization 3 years later [9]. In contrast, another study found that teacher–student relationships among students in grades four to six did not significantly predict bullying victimization [10]. These mixed findings underline the need for further research, particularly on how teacher–student relationships influence bullying in early adolescence, a period characterized by significant shifts in both social and cognitive development. In the Chinese context, where teacher authority is emphasized, these relationships may play a unique role in shaping adolescents’ experiences with bullying. Therefore, it is crucial to understand how teacher support and authoritative classroom environments impact adolescents’ vulnerability to bullying in the context of Chinese educational pressures.

While some studies have examined the relationships among perceived control, teacher–student relationships, and bullying victimization, the underlying interaction mechanisms among these factors remain underexplored. Additionally, these factors change over time as adolescents mature [11,12,13]. However, it is still unclear how the relationship between perceived control, teacher–student relationships, and bullying victimization evolves over the course of early adolescence. To address this gap, the present study aims to explore the developmental trajectories of these factors over one year using the latent growth model (LGM). Furthermore, this study will examine the longitudinal mediating role of teacher–student relationships in the relationship between perceived control and bullying victimization. This research is particularly significant in the Chinese context, where high academic pressures and teacher authority may influence the development of perceived control and the dynamics of bullying victimization.

### 1.1. Perceived Control and Bullying Victimization

Perceived control refers to an individual’s belief in their ability to influence their environment and regulate their behavior [14]. It encompasses a sense of agency and mastery over external circumstances, which plays a vital role in psychological well-being. Adolescents with higher perceived control are better equipped to cope with challenges, leading to improved mental health, greater resilience, and reduced negative emotional responses to stressors [11].

Adolescents who experience bullying may develop anxiety, depression, and social withdrawal. Perceived control has been identified as an important factor in reducing the risk of bullying. According to the Conservation of Resources (COR) model, individuals with low perceived control lack critical resources, such as effective coping strategies and social support, making them more vulnerable to bullying [15]. Research shows that low perceived control is a risk factor for bullying victimization [16]. For example, a longitudinal study found that adolescents with lower perceived control are more likely to experience bullying over time [17]. This suggests that perceived control affects both immediate and long-term vulnerability to bullying. Adolescents with higher perceived control are more proactive in seeking solutions independently, rather than relying on external intervention [18]. This sense of autonomy enables them to regain control and mitigate the harmful effects of bullying [19].

Despite the established link between perceived control and bullying, longitudinal studies on how perceived control protects against bullying over time are limited. Research suggests that perceived control typically increases as adolescents grow, which may reduce issues like depression, anxiety, and aggression related to bullying [20]. However, it remains unclear how perceived control and bullying victimization change together over time.

To address these questions, it is crucial to examine both the average levels (intercepts) and changes over time (slopes) of perceived control and bullying. The intercept reflects the initial level of perceived control or bullying, while the slope indicates the rate of change. By studying these factors, researchers can better understand whether adolescents with lower initial perceived control are more likely to experience higher rates of bullying, and whether increases in perceived control are associated with reductions in bullying. This analysis can provide valuable insights for identifying effective intervention strategies during adolescence.

### 1.2. Perceived Control and Teacher–Student Relationships

According to the locus of control theory, individuals are categorized as either internal or external in terms of their perceived control [21]. Adolescents with a strong perceived control are typically internals, who trust that their actions can shape their academic achievements and interpersonal relationships. Perceived control motivates adolescents to take initiative and engage proactively with their teachers, fostering positive, supportive interactions [22]. In contrast, adolescents with diminished perceived control tend to exhibit higher incidence of negative behaviors [23], struggle to receive positive feedback from teachers, and have difficulty developing supportive teacher–student relationships [24]. Although few studies have directly explored the impact of perceived control on teacher–student relationships, the existing literature suggests that perceived control may play a facilitative role in fostering teacher–student relationships. According to Self-Determination Theory, adolescents with higher perceived control are more likely to experience autonomy and relatedness, both of which are critical for developing positive teacher–student relationships [25]. Adolescents with higher perceived control engage more actively in classroom interactions and proactively seek help from their teachers [26]. Adolescent students with perceived control believe that they are valued and supported by their teachers, and they possess the confidence to strengthen these relationships [27]. Relatedness is a fundamental human need [28]. Teachers, as significant figures within the school environment, play a crucial role in facilitating academic engagement, emotional support, and positive social interactions for adolescents. Given adolescents’ inherent need for relatedness, adolescents naturally seek to establish meaningful and supportive emotional connections with their teachers. In supportive school environments, teachers can fulfill this need, helping students build a sense of belonging and improving their overall social and academic outcomes. Thus, this study seeks to explore how perceived control can enhances teacher–student relationships, particularly by examining its influence on adolescent–teacher interactions.

### 1.3. Teacher–Student Relationships and Bullying Victimization

Teachers play a central role in the preventing and intervening of bullying victimization [29]. Grounded in attachment theory, teachers become significant adults in the lives of adolescents at school, partially assuming some of the roles of parents [30]. As attachment figures, teachers provide a sense of security, which students may seek during times of distress, serving as a stable base for social exploration. This sense of security helps foster peer relationships and mitigate bullying [31]. In China, where academic pressures are high, teachers also provide a sense of stability amidst the emotional challenges adolescents face due to academic and social expectations.

The quality of teacher–student relationships has been shown to correlate with levels of peer aggression. Positive teacher–student relationships are often associated with lower levels of aggression, while negative relationships may exacerbate being a victim of bullying [32]. Teacher support is a strong predictor of whether a student is likely to be bullied. For example, perceived teacher care and support can significantly reduce the likelihood of bullying victimization [33]. However, it is still unclear whether these effects persist over time, particularly in culturally distinct educational settings like China, where teacher authority and academic pressures may influence the dynamics of teacher–student relationships.

Previous research on the longitudinal impact of teacher–student relationships has yielded mixed findings. A cross-lagged study found that supportive teacher–student relationships predicted reduced victimization over time [32]. In contrast, other studies found no significant longitudinal impact, suggesting that teacher–student relationships might not consistently influence peer aggression [34]. Furthermore, teacher favoritism can lead to jealousy among peers, paradoxically increasing bullying victimization despite positive teacher–student relationships [35]. In China, where teacher authority is particularly strong, favoritism can create social divisions among students, exacerbating the risk of bullying. These conflicting findings suggest that the influence of teacher–student relationships on bullying victimization is complex and warrants further exploration across different cultural and developmental contexts. In the Chinese educational environment, where academic performance plays a significant role in social dynamics, teacher–student relationships may have unique implications for bullying victimization.

Many studies have utilized cross-lagged models to explore the relationship between teacher–student relationships and bullying [32,33,34]. While these models provide insights into the direction of causality, they do not adequately address the specific roles of initial levels and growth rates of key variables. This gap in the literature highlights the need for a more nuanced understanding of how teacher–student relationships evolve over time and their differential effects on bullying. Specifically, initial levels of teacher support and the growth trajectory of these relationships may exert distinct influences, potentially explaining inconsistencies in prior findings.

Therefore, this study aims to explore the impact of teacher–student relationships on bullying victimization by analyzing both initial levels and growth rates using latent growth models. This approach will provide a deeper understanding of how the dynamics of teacher–student relationships contribute to bullying prevention and how these influences evolve over time, particularly in the context of China’s academic pressures and teacher authority.

### 1.4. The Current Research

In summary, our study fills a gap in the literature by examining the longitudinal change trajectories of perceived control, teacher–student relationships, and bullying victimization among Chinese early adolescents. It specifically aims to investigate how teacher–student relationships mediate the relationship between perceived control and bullying victimization over time. The hypotheses are as follows: (1) Bullying victimization in early adolescents will decrease over time, while perceived control and teacher–student relationships will increase. (2) Changes in perceived control will directly predict changes in bullying victimization, and this relationship will be mediated by teacher–student relationships. (3) The initial level of teacher–student relationships positively predicts adolescents’ experiences of bullying victimization. (4) Over the long term, teacher–student relationships will negatively predict bullying victimization in early adolescents.

## 2. Methods

### 2.1. Participants and Procedure

A total of 1454 adolescents from the seventh and eighth grades of a middle school were selected for a three-wave longitudinal study, with assessments conducted every six months over a year-long period. The number of participants was 1414 (*Mage* = 13.63 years, *SD* = 0.76, June 2022), 1328 (*Mage* = 14.04 years, *SD* = 0.73, December 2022), and 1246 (*Mage* = 14.66 years, *SD* = 1.14, June 2023), respectively, for each wave. A total of 208 students were unable to complete the full assessment due to illness, temporary withdrawal, transfer, or other related reasons. Little’s MCAR test was applied to missing data for the study variables, indicating that the missing data pattern was completely random (χ^2^ = 33.20, *p* = 0.31). Therefore, Full Information Maximum Likelihood Estimation was employed to address the missing data. A total of 1246 participants (*Mage* = 13.58 years, *SD* = 0.75) who completed all three assessments were retained, including 601 males (48.23%) and 645 females (51.77%).

This study was approved by the Ethics Committee of the first author’s University (Ethical Approval No. ZZUIRB2020-021). Consent was obtained from both parents and students prior to the survey. At each wave, questionnaires were administered in classrooms during regular class sessions, with students completing them independently under the guidance of two trained graduate students in psychology. The students were supervised during the process, which lasted no more than 30 min. Participants were informed that they could withdraw from the study at any time without penalty. All collected data were directly encrypted and stored by the first author.

### 2.2. Measurement Instruments

#### 2.2.1. Bullying Victimization

Bullying victimization was assessed using the Chinese version of the Delaware Bullying Victimization Scale (DBVS). The original DBVS was developed in English by Bear [36], and the revised version was validated for use among Chinese adolescents by Xie [37]. It consists of 17 items, scoring on a 6-point Likert scale ranging from 1 (never) to 6 (every day). A higher total score indicates a more severe level of bullying victimization. The scale demonstrated good internal consistency across the three waves of testing, with Cronbach’s α coefficients of 0.91, 0.94, and 0.94, respectively.

#### 2.2.2. Perceived Control

The Sense of Control Scale (SCS) was used to assess adolescents’ sense of personal control and perceived constraints. The original SCS was developed in English by Lachman [38], and the revised version was validated for use among Chinese adolescents by Li [39]. It consists of 12 items scored on a 7-point Likert scale (1 = strongly disagree, 7 = strongly agree). A higher total score indicates stronger perceived control. In this study, the Cronbach’s α reliability coefficients of the scale at the three time points were 0.72, 0.80, and 0.82, respectively.

#### 2.2.3. Teacher–Student Relationship

The Teacher–Student Relationship Scale (TSRS) was employed for assessment, with students reporting their responses through self-assessment. The original TSRS was developed in English by Pianta [40], and the revised version was adapted and validated for use among Chinese adolescents by Yuan [41]. This scale includes dimensions of closeness, conflict (reverse-scored), support, and satisfaction, comprising 23 items scored on a 5-point Likert scale ranging from 1 (totally disagree) to 5 (totally agree). A higher total score indicates a better teacher–student relationship. In this study, the Cronbach’s α reliability coefficients of the scale at the three time points were 0.73, 0.74, and 0.74, respectively.

### 2.3. Data Analysis

SPSS 25.0 was utilized for reliability, descriptive, and correlational analyses. Latent growth models were constructed using Mplus Version 8.3. Analyses were performed in the following steps. Initially, linear unconditional latent growth models will be constructed for bullying victimization, perceived control, and teacher–student relationships, respectively. For these models, the latent intercept factor across the three time points will be set to 1, 1, 1, and the latent slope factor will be set to 0, 1, 2, in order to explore the developmental trajectories of each variable. Subsequently, a latent growth mediation model will be established to examine the longitudinal mediation mechanism between the initial levels and growth rates of these variables. The Bootstrap method (with 5000 resamples) will be used to estimate the regression coefficients and their 95% confidence intervals. If the confidence interval does not contain zero, it indicates that the regression coefficient is statistically significant. Gender (0 = male, 1 = female) was incorporated as a control variable in all models. Model fit will be evaluated using the criteria of *χ*^2^/*df* < 5, CFI/TLI > 0.90, and RMSEA/SRMR < 0.08 for good model fitting.

## 3. Results

### 3.1. Descriptive Statistics and Correlational Analysis

The mean, standard deviation, and correlation matrix of bullying victimization, perceived control, and teacher–student relationships across three assessments are presented in Table 1. Bullying victimization from T1 to T3 was found to be significantly negatively correlated with perceived control from T1 to T3 (*r* = −0.16 to −0.33, *p* < 0.001), as well as with teacher–student relationship from T1 to T3 (*r* = −0.07 to −0.20, *p* < 0.001). Furthermore, a significant positive correlation was observed between perceived control and teacher–student relationship from T1 to T3 (*r* = 0.18 to 0.42, *p* < 0.001).

### 3.2. Trajectories of Bullying Victimization, Perceived Control, and Teacher–Student Relationships Development

Unconditional linear growth models for bullying victimization, perceived control, and teacher–student relationships were constructed separately. The results indicate (see Table 2) that the unconditional linear growth model for bullying victimization fits well, indicated by *χ*^2^/*df* = 4.37, *p* < 0.05, CFI = 0.98, TLI = 0.92, RMSEA = 0.05, SRMR = 0.02. The intercept, representing the initial level of bullying victimization, was 21.29 (*p* < 0.001), with a slope mean of −0.69 (*p* < 0.001), indicating a significant decreasing trend in bullying victimization among early adolescents over time. The variance of the intercept was 36.4 (*p* < 0.001), and the variance of the slope was 6.69 (*p* > 0.05), suggesting individual differences in the initial levels of bullying victimization, but not in the rates of development. The correlation between the intercept and slope was not significant (*r* = −0.15, *p* > 0.05), indicating that the initial level of bullying victimization was not strongly related to its rate of change. After incorporating the control variables, the model fit remains satisfactory, with *χ*^2^/*df* = 2.16, *p* = 0.12, CFI = 0.97, TLI = 0.92, RMSEA = 0.03, and SRMR = 0.08. Gender had no statistically significant effect on the intercept of bullying victimization (*β* = 0.014, *p* = 0.76). Gender had a significant positive effect on the slope of bullying victimization (*β* = 0.130, *p* < 0.001), suggesting that females experience a higher rate of decrease in bullying victimization compared to males.

The unconditional linear growth model for perceived control also demonstrated a good fit, with *χ*^2^/*df* = 0.81, *p* = 0.37, CFI = 1.00, TLI = 1.00, RMSEA < 0.01, and SRMR = 0.01. The intercept for perceived control, representing the initial level, was 54.73 (*p* < 0.001), with a slope mean of 0.57 (*p* < 0.001), indicating that early adolescents’ perceived control started from a high level and significantly increased over time. The variance of the intercept was 47.86 (*p* < 0.001), and the variance of the slope was 15.68 (*p* < 0.001), indicating individual differences in both the initial levels and the rates of development for perceived control. The correlation between the intercept and slope was not significant (*r* = −0.26, *p* > 0.05), indicating that the initial level of perceived control was not strongly related to its rate of change. After incorporating the control variables, the model fit remains satisfactory, with *χ*^2^/*df* = 1.97, *p* = 0.14, CFI = 0.98, TLI = 0.93, RMSEA = 0.03, and SRMR = 0.05. Gender had a significant negative effect on the intercept of perceived control (*β* = −0.24, *p* < 0.001), indicating that females have a lower initial level of perceived control compared to males. Gender had a significant positive effect on the slope of perceived control (*β* = 0.12, *p* < 0.001), suggesting that females have a higher rate of increase in perceived control compared to males.

The unconditional linear growth model for teacher–student relationships also fitted well, with *χ*^2^/*df* = 6.16, *p* < 0.05, CFI = 0.98, TLI = 0.95, RMSEA = 0.07, and SRMR = 0.02. The intercept for teacher–student relationships, indicating the initial level, was 68.98 (*p* < 0.001), with a slope mean of 0.98 (*p* < 0.001), signifying a significant upward trend in teacher–student relationships among early adolescents over time. The variance of the intercept was 43.94 (*p* < 0.001), and the variance of the slope was 8.64 (*p* < 0.001), indicating individual differences in both the initial levels and the rates of development for teacher–student relationships. The correlation between the intercept and slope was significant (*r* = −0.31, *p* < 0.01), suggesting that a higher initial level of teacher–student relationships was associated with a slower rate of improvement in teacher–student relationships. After incorporating the control variables, the model fit significantly declined, with *χ*^2^/*df* = 22.61, *p* < 0.05, CFI = 0.90, TLI = 0.71, RMSEA = 0.13, and SRMR = 0.06. This decline suggests that including gender as a control variable, due to its weak correlation with the dependent variable, added noise to the model and reduced its explanatory power.

### 3.3. The Longitudinal Mediating Role of Teacher–Student Relationships Between Perceived Control and Bullying Victimization

A latent variable growth model was employed to construct a mediation model of intercepts and slopes, exploring the relationship between the developmental trends of perceived control, teacher–student relationships, and bullying victimization. The results (Figure 1) demonstrated a good model fit, with *χ*^2^/*df* = 6.05, *p* < 0.05, CFI = 0.96, TLI = 0.94, RMSEA = 0.07, and SRMR = 0.04. After incorporating the control variables, the model fit significantly declined, with *χ*^2^/*df* = 18.68, *p* < 0.05, CFI = 0.88, TLI = 0.79, RMSEA = 0.12, and SRMR = 0.07. This suggests that the inclusion of gender, which showed a weak correlation with the dependent variable, may have contributed little to the model and introduced unnecessary complexity, leading to a poorer fit.

The intercept of perceived control positively predicted the intercept of teacher–student relationships (*β* = 0.63, 95%CI [0.56, 0.70]). It also negatively predicted the intercept of bullying victimization (*β* = −0.97, 95%CI [−1.26, −0.74]). This means that higher initial perceived control led to better teacher–student relationships and less bullying victimization. The intercept of teacher–student relationships, however, positively predicted bullying victimization (*β* = 0.32, 95%CI [0.14, 0.63]). This suggests that better initial relationships could also increase the likelihood of bullying, with an indirect effect of 0.2. Overall, perceived control significantly reduced bullying victimization, with a net effect of −0.77. This effect was partially mediated by teacher–student relationships. The slope of perceived control did not directly predict the slope of bullying victimization (*β* = 0.13, 95%CI [−0.60, 0.48]). However, it indirectly influenced it through the slope of teacher–student relationships (*β* = −0.23, 95%CI [−0.60, −0.10]). This suggests that changes in perceived control influenced bullying by affecting changes in teacher–student relationships (see Table 3).

To explore the mediating role of teacher–student relationships further, Bootstrap resampling with 5000 samples was used. Initial perceived control had a significant direct effect on bullying victimization. Its indirect effect was 0.20, which accounted for 17.09% of the total effect. This shows that teacher–student relationships played a partially mediating role. The direct effect of changes in perceived control on bullying victimization was not significant. However, the indirect effect was 0.23. This means that the change in teacher–student relationships fully mediated the effect of changes in perceived control on bullying.

## 4. Discussion

This study, conducted over one year with three waves of longitudinal tracking, found that perceived control and teacher–student relationships showed a linear increase over time, while bullying victimization decreased. The initial level of perceived control significantly predicted the initial level of bullying victimization and indirectly influenced it through the initial level of teacher–student relationships. The increasing trend in perceived control also indirectly contributed to the decreasing trend in bullying victimization, but only through improving teacher–student relationships. This suggests that adolescents with initially higher perceived control are less likely to experience bullying victimization. However, better teacher–student relationships at the outset were associated with a higher risk of bullying. Over time, adolescents’ perceived control strengthens, leading to improved teacher–student relationships, which in turn further reduces the likelihood of bullying victimization.

### 4.1. The Developmental Trajectories of Early Adolescent Perceived Control, Teacher–Student Relationships, and Bullying Victimization

This study found that perceived control exhibited a linear increasing trend over time, suggesting that adolescents’ perceived control strengthens as they progress through early adolescence. This finding can be understood from a developmental perspective, considering the significant cognitive, emotional, and social changes that occur during this stage of life [2]. As adolescents’ self-awareness matures, they develop a better understanding of their surroundings and the factors influencing their lives, which in turn enhances their perception of control [42]. Cognitive advances, such as improved executive functioning and decision-making abilities, enable adolescents to plan, strategize, and evaluate their actions more effectively, which reinforces their sense of control [43]. Social influences also contribute to the increase in perceived control. During adolescence, societal expectations encourage young people to take greater responsibility for their own actions. As adolescents face increased pressure to become more independent, they are motivated to adopt proactive behaviors, which further enhances their perceived control [44]. Moreover, positive school experiences and supportive peer interactions contribute to adolescents’ confidence in their ability to influence outcomes in both academic and social contexts. Together, these factors support growth in perceived control during adolescence.

Teacher–student relationships also showed a linear increasing trend over time, suggesting that these relationships tend to become more supportive and constructive as adolescents mature. This finding is consistent with prior research, which highlights the evolving nature of teacher–student interactions, especially as students transition into middle school [45,46]. During this period, adolescents face increased academic challenges, prompting them to rely more on teachers for both academic and emotional support. Teachers often take on quasi-parental roles, providing guidance and emotional reassurance, which helps adolescents navigate the complexities of this developmental stage [30]. As adolescents’ social and emotional skills improve, they tend to place more value on these teacher–student relationships, recognizing teachers as critical sources of support. This shift reflects adolescents’ growing understanding of authority figures as sources of guidance, rather than merely enforcers of rules, which may foster academic engagement and emotional resilience [47].

Lastly, bullying victimization among early adolescents demonstrated a linear decreasing trend over time, in contrast to the increases observed in perceived control and teacher–student relationships. This decline in victimization can be attributed to several developmental factors. As adolescents mature, cognitive changes such as improved empathy and perspective-taking reduce the likelihood of bullying. Adolescents’ growing ability to understand the emotions and perspectives of others decreases their tendency to engage in aggressive behaviors [48]. Additionally, the development of social problem-solving skills, often promoted through peer interactions and social learning, provides adolescents with more effective ways to resolve conflicts without resorting to aggression [49]. As peer relationships become more central, adolescents prioritize maintaining harmony, which discourages aggressive behaviors. Positive peer interactions encourage pro-social behaviors, which help create a supportive social environment that reduces aggressive behaviors [50].

### 4.2. The Longitudinal Impact of Perceived Control on Bullying Victimization

Employing latent growth models, it was observed that, in early adolescence, the initial level of perceived control negatively predicted the initial level of bullying victimization, aligning with previous research findings [16,50,51,52,53]. Early adolescents who perceive a lack of control over their lives and studies often exhibit increased feelings of helplessness, hopelessness, and depression [54]. These emotional states can render them more vulnerable to victimization, as they may struggle to effectively manage stressful situations and defend themselves against aggression [55]. Adolescents with a stronger sense of perceived control typically exhibit higher self-esteem, confidence, and coping strategies, enabling them to confront challenges more effectively, thereby reducing their susceptibility to bullying victimization [56,57]. However, this study also found that an increasing trend in perceived control did not directly predict a decreasing trend in bullying victimization, which suggests that the relationship between these variables may not be as straightforward as previously assumed. This discrepancy could be attributed to the complex interaction between individual growth and environmental factors. Specifically, in the collectivistic context of Chinese society, adolescents may prioritize social harmony and group conformity over assertive behaviors. Despite an increase in perceived control, adolescents may be less inclined to confront bullying directly or assertively due to cultural norms that emphasize fitting in, avoiding conflict, and maintaining social cohesion within peer groups [58]. Thus, while increasing perceived control may enhance adolescents’ ability to cope with stress and challenges, it may not automatically translate into a reduction in bullying victimization in contexts where social harmony and conformity are prioritized over individual assertion.

### 4.3. The Longitudinal Mediating Role of Teacher–Student Relationships Between Perceived Control and Bullying Victimization

This study found that the initial level of perceived control positively predicted the initial level of teacher–student relationships, and the developmental trend of perceived control positively predicted the developmental trend of teacher–student relationships, consistent with previous research [59]. Adolescents with lower perceived control may have experienced heightened negative affect, including feelings of anger and anxiety, which can accumulate over time and manifest as frustration or maladaptive behaviors [60]. These behaviors may, in turn, erode the quality of their interactions with teachers, positioning teacher–student relationships as a mediating variable influenced by changes in perceived control and potentially shaping subsequent behavioral outcomes [61].

Interestingly, this study found that the initial quality of teacher–student relationships positively predicted the initial level of bullying victimization. This finding suggests that students with stronger initial relationships with teachers may experience increased peer victimization, potentially due to social dynamics such as peer jealousy or exclusion. This finding aligns with the observations by Bokkel et al. [35] and Demol et al. [32], who noted that students perceived as having preferential relationships with teachers can become targets of peer hostility in competitive or high-pressure educational settings. Within such contexts, forming strong bonds with teachers may inadvertently distinguish these students from their peers, leading to social tension and increased bullying risk. Thus, while supportive teacher–student relationships offer critical emotional and academic benefits, they may also generate unintended social consequences in peer interactions. In terms of rate of change, an increase in teacher–student relationships was associated with a decline in bullying victimization rates. Adolescents with high-quality teacher–student relationships are more likely to develop adaptive conflict resolution skills and benefit from consistent emotional support, thereby mitigating experiences of victimization [62]. Close relationships with teachers can foster a sense of safety and trust, enhancing students’ willingness to disclose bullying experiences and seek assistance [31,63]. This, in turn, allows teachers to recognize and address victimization more effectively, providing timely intervention and emotional support [62,64].

The findings of the longitudinal mediation model indicate that perceived control exerts an indirect influence on bullying victimization through its impact on teacher–student relationships. Adolescents with a stronger sense of perceived control are more likely to establish positive and supportive relationships with their teachers, which serve as protective factors against bullying. This underscores the importance of interventions aimed at fostering adolescents’ perceived control, as such efforts can strengthen teacher–student bonds and reduce the likelihood of victimization. Adolescents with high perceived control are generally better equipped to manage stressors, engage in positive social interactions, and develop resilience, thereby decreasing their vulnerability to bullying [65]. Strengthening teacher–student relationships thus emerges as a critical component in helping adolescents navigate stress, enhance their coping capacity, and ultimately reduce bullying victimization.

## 5. Research Limitations, Practical Implications, and Future Directions

This study contributes to understanding bullying victimization among adolescents by identifying patterns of change and highlighting key risk factors. The findings offer practical implications for educational practice and policy [66,67]. First, the positive association between perceived control and reduced bullying victimization underscores the importance of school programs that enhance students’ sense of agency and self-efficacy, such as social–emotional learning (SEL) programs aimed at building resilience and coping skills [66]. Second, the mediating role of teacher–student relationships suggests that fostering positive interactions between teachers and students is crucial. Training teachers to cultivate empathetic, supportive connections with students can create a protective school climate and encourage students to seek help when facing bullying [67]. Additionally, recognizing that strong teacher–student bonds may lead to social tension among peers highlights the need for strategies that build inclusive peer environments, promoting cohesion and reducing jealousy-driven dynamics [67].

However, several limitations should be noted. First, this study collected only three waves of data, limiting the analysis to a linear growth model. Additional data waves could help explore nonlinear growth trajectories, providing deeper insights into how bullying evolves over time. Second, this study takes a variable-centered approach, which overlooks the heterogeneity among adolescents experiencing bullying. This limits our understanding of distinct subgroups and their unique patterns of victimization and resilience. Future research could adopt an individual-centered approach to better capture these nuances. Lastly, this study does not consider contextual factors like school climate or family environment, which significantly influence bullying experiences. Including these factors in future research would provide a more comprehensive understanding of bullying dynamics and lead to more effective interventions. Future research should incorporate nonlinear growth models, individual-centered analyses, and contextual factors to address these gaps and provide a more complete picture of adolescent bullying.

## Figures and Tables

**Figure 1 behavsci-14-01198-f001:**
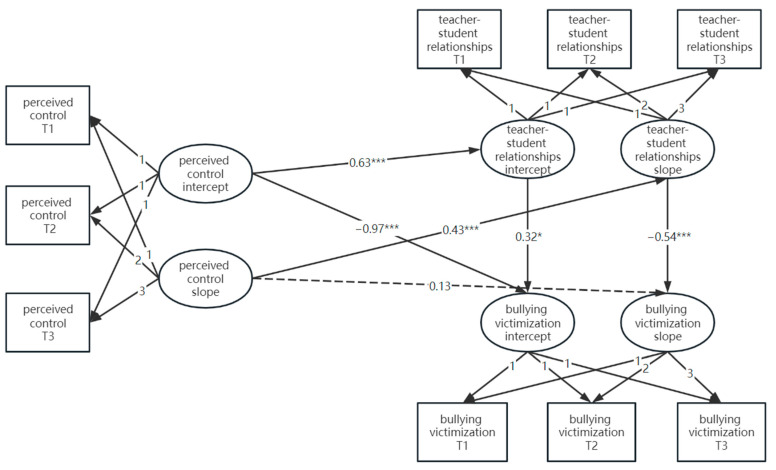
Latent growth mediation model. Note. * *p* < 0.05, *** *p* < 0.001.

**Table 1 behavsci-14-01198-t001:** The mean, standard deviation, and correlation matrix for each variable.

Variable	*M*	*SD*	1	2	3	4	5	6	7	8
1. bullying victimization T1	21.83	8.28								
2. bullying victimization T2	19.67	7.31	0.46 ***							
3. bullying victimization T3	19.23	6.82	0.36 ***	0.46 ***						
4. perceived control T1	55.34	9.09	−0.33 ***	−0.21 ***	−0.16 ***					
5. perceived control T2	55.72	9.74	−0.31 ***	−0.28 ***	−0.28 ***	0.56 ***				
6. perceived control T3	55.92	9.69	−0.28 ***	−0.24 ***	−0.27 ***	0.46 ***	0.60 ***			
7. teacher–student relationships T1	68.77	9.52	−0.20 ***	−0.12 ***	−0.07 ***	0.31 ***	0.24 ***	0.23 ***		
8. teacher–student relationships T2	70.41	9.27	−0.13 ***	−0.19 ***	−0.14 ***	0.22 ***	0.32 ***	0.32 ***	0.43 ***	
9. teacher–student relationships T3	71.22	9.22	−0.17 ***	−0.17 ***	−0.14 ***	0.18 ***	0.26 ***	0.42 ***	0.36 ***	0.51 ***

Note. *** *p* < 0.001.

**Table 2 behavsci-14-01198-t002:** Unconditional linear growth model for each variable.

	Intercept		Slope		*r* (Intercept with Slope)	Model Fit			
	Mean	Variance	Mean	Variance		CFI	TLI	RMSEA	SRMR
bullying victimization	21.29 ***	36.40 ***	−0.69 ***	6.69	−0.15	0.98	0.92	0.05	0.02
perceived control	54.73 ***	47.86 ***	0.57 ***	15.68 ***	−0.26	1.00	1.00	<0.01	0.01
teacher–student relationships	68.98 ***	43.94 ***	0.98 ***	8.64 ***	−0.31 **	0.98	0.95	0.07	0.02

Note. ** *p* < 0.01, *** *p* < 0.001.

**Table 3 behavsci-14-01198-t003:** Analysis of mediating effects.

Mediating Effect	*β*	*SE*	95% Confidence Interval
Intercept: Perceived control → Teacher–student relationships → Bullying victimization	0.20	0.10	[0.07, 0.45]
Slope: Perceived control → Teacher–student relationships → Bullying victimization	−0.23	0.52	[−0.60, −0.10]

## Data Availability

The datasets generated and/or analyzed during the current study are not publicly available but are available from the corresponding author on reasonable request.

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
