# Peer review of "Breaking the Cycle: Perceived Control and Teacher–Student Relationships Shield Adolescents from Bullying Victimization over Time"

_behavsci, 2024, doi:10.3390/bs14121198_

Round 1

Reviewer 1 Report

Comments and Suggestions for Authors

As this dataset is from China, context must be provided to the Chinese culture and school system, both in the front end and in the discussion.

Why are are there are no control variables?

Comments on the Quality of English Language

This article was difficult to read in many instances. Sentences were duplicated entirely or concepts were discussed multiple times in a short space. I believe this is an interesting article with innovative data and statistics; however, the language must be reassessed and condensed throughout.

Author Response

Dear Reviewer,

Thank you for your thoughtful and constructive comments on our manuscript. We greatly appreciate the time and effort you took to review our paper and provide such valuable feedback.

Based on your suggestions, we have made several key revisions to improve the manuscript. First, we have added more context regarding the Chinese culture and education system in both the Introduction and Discussion sections, which provides readers with a better understanding of the cultural and educational pressures that adolescents in China face, particularly in relation to bullying and other school-related challenges. Second, in response to your comment about control variables, we have incorporated gender as a control variable in our models and updated the manuscript to reflect its impact on bullying victimization, perceived control, teacher-student relationships, and the mediation model. This ensures that we account for important factors influencing the outcomes. Lastly, we have carefully reviewed and revised the manuscript to improve its readability, eliminate redundancy, and make the presentation of our findings clearer and more concise.

Comments 1: As this dataset is from China, context must be provided to the Chinese culture and school system, both in the front end and in the discussion.

Response 1:

Thank you for pointing this out. I agree with this comment. Therefore, I have provided additional context regarding Chinese culture and the Chinese education system in the Introduction and discussion section to help readers understand the unique educational pressures and cultural factors that influence adolescents in China.

The revisions can be found on page 1-4, page 9-11 of the revised manuscript.

Updated text in the manuscript:

  1. Introduction

Bullying victimization refers to the phenomenon wherein an individual is subjected to prolonged or repetitive mistreatment or harm by one or more peers [1]. Early adolescence is a period of rapid physical and mental growth, making individuals particularly sensitive to social influences [2]. During this period, adolescents heavily rely on social relationships for validation and a sense of belonging, and undergo intense self-exploration [3]. When these needs for positive social connection are not fulfilled or when the process of self-exploration encounters setbacks, early adolescents may become more vulnerable to negative social experiences such as bullying [3]. This developmental stage also coincides with a peak in the prevalence of bullying behaviors, with approximately 30.4% of early adolescents experiencing bullying [4, 5]. Especially, Chinese adolescence coincides with significant academic pressures, particularly the preparation for the Zhongkao (high school entrance exam). The competitive nature of the Chinese educational system, where academic performance is highly valued, often leads to increased social comparison among peers. Adolescents who do not perform as well academically are more likely to feel socially isolated, stigmatized, and may become targets of bullying by their peers. Bullying can lead to a range of adverse outcomes, including increased risks of depression, anxiety, social withdrawal, and even suicidal ideation [6,7]. Given these serious consequences, bullying victimization represents a critical social issue that demands comprehensive intervention and prevention efforts.

Existing research indicates that perceived control and teacher-student relationships may serve as effective predictive factors for bullying victimization [8,9]. While longitudinal studies on childhood have explored the role of perceived control in bullying victimization, research on early adolescence remains limited. For instance, children's perceived control at the age of 5 could significantly predict their experience of bullying victimization at the age of 6 [8]. However, there is a gap in similar research focusing on early adolescents, who differ from children in their social needs, cognitive abilities, and response to external stressors. These developmental differences likely influence how perceived control impacts bullying dynamics. Therefore, it is essential to investigate the role of perceived control in early adolescence, particularly within the context of the increasing academic and social pressures that shape this stage of development.

The longitudinal relationship between teacher-student relationships and bullying victimization remains inconclusive. For example, teacher-student relationships among 13-year-old adolescents could predict their experience of bullying victimization three years later [9]. In contrast, another study found that teacher-student relationships among students in grades four to six did not significantly predict bullying victimization [10]. These mixed findings underline the need for further research, particularly on how teacher-student relationships influence bullying in early adolescence, a period characterized by significant shifts in both social and cognitive development. In the Chinese context, where teacher authority is emphasized, these relationships may play a unique role in shaping adolescents' experiences with bullying. Therefore, it is crucial to understand how teacher support and authoritative classroom environments impact adolescents' vulnerability to bullying in the context of Chinese educational pressures.

While some studies have examined the relationships among perceived control, teacher-student relationships, and bullying victimization, the underlying interaction mechanisms among these factors remain underexplored. Additionally, these factors change over time as adolescents mature [11-13]. However, it is still unclear how the relationship between perceived control, teacher-student relationships, and bullying victimization evolves over the course of early adolescence. To address this gap, the present study aims to explore the developmental trajectories of these factors over one year using the Latent Growth Model (LGM). Furthermore, the study will examine the longitudinal mediating role of teacher-student relationships in the relationship between perceived control and bullying victimization. This research is particularly significant in the Chinese context, where high academic pressures and teacher authority may influence the development of perceived control and the dynamics of bullying victimization.

1.1. Perceived Control and Bullying Victimization

Perceived control refers to an individual's belief in their ability to influence their environment and regulate their behavior [14]. It encompasses a sense of agency and mastery over external circumstances, which plays a vital role in psychological well-being. Adolescents with higher perceived control are better equipped to cope with challenges, leading to improved mental health, greater resilience, and reduced negative emotional responses to stressors [11].

Adolescents who experience bullying may develop anxiety, depression, and social withdrawal. Perceived control has been identified as an important factor in reducing the risk of bullying. According to the Conservation of Resources (COR) model, individuals with low perceived control lack critical resources, such as effective coping strategies and social support, making them more vulnerable to bullying [15]. Research shows that low perceived control is a risk factor for bullying victimization [16]. For example, a longitudinal study found that adolescents with lower perceived control are more likely to experience bullying over time [17]. This suggests that perceived control affects both immediate and long-term vulnerability to bullying. Adolescents with higher perceived control are more proactive in seeking solutions independently, rather than relying on external intervention [18]. This sense of autonomy enables them to regain control and mitigate the harmful effects of bullying [19].

Despite the established link between perceived control and bullying, longitudinal studies on how perceived control protects against bullying over time are limited. Research suggests that perceived control typically increases as adolescents grow, which may reduce issues like depression, anxiety, and aggression related to bullying [20]. However, it remains unclear how perceived control and bullying victimization change together over time.

To address these questions, it is crucial to examine both the average levels (intercepts) and changes over time (slopes) of perceived control and bullying. The intercept reflects the initial level of perceived control or bullying, while the slope indicates the rate of change. By studying these factors, researchers can better understand whether adolescents with lower initial perceived control are more likely to experience higher rates of bullying, and whether increases in perceived control are associated with reductions in bullying. This analysis can provide valuable insights for identifying effective intervention strategies during adolescence.

1.2. Perceived Control and Teacher-Student Relationships

According to Locus of Control theory, individuals are categorized as either internal or external in terms of their perceived control [21]. Adolescents with a strong perceived control are typically internals, who trust that their actions can shape their academic achievements and interpersonal relationships. Perceived control motivates adolescents to take initiative and engage proactively with their teachers, fostering positive, supportive interactions [22]. In contrast, adolescents with diminished perceived control tend to exhibit higher incidence of negative behaviors [23], struggle to receive positive feedback from teachers, and have difficulty developing supportive teacher-student relationships [24]. Although few studies have directly explored the impact of perceived control on teacher-student relationships, existing literature suggests that perceived control may play a facilitative role in fostering teacher-student relationships. According to Self-Determination Theory, adolescents with higher perceived control are more likely to experience autonomy and relatedness, both of which are critical for developing positive teacher-student relationships [25]. Adolescents with higher perceived control engage more actively in classroom interactions and proactively seek help from their teachers [26]. Adolescent students with perceived control believe that they are valued and supported by their teachers, and they possess the confidence to strengthen these relationships [27]. Relatedness is a fundamental human need [28]. Teachers, as significant figures within the school environment, play a crucial role in facilitating academic engagement, emotional support, and positive social interactions for adolescents. Given adolescents' inherent need for relatedness, adolescents naturally seek to establish meaningful and supportive emotional connections with their teachers. In supportive school environments, teachers can fulfill this need, helping students build a sense of belonging and improving their overall social and academic outcomes. Thus, this study seeks to explore how perceived control can enhances teacher-student relationships, particularly by examining its influence on adolescent-teacher interactions.

1.3. Teacher-Student Relationships and Bullying Victimization

Teachers play a central role in the preventing and intervening of bullying victimization [29]. Grounded in attachment theory, teachers become significant adults in the lives of adolescents at school, partially assuming some of the roles of parents [30]. As attachment figures, teachers provide a sense of security, which students may seek during times of distress, serving as a stable base for social exploration. This sense of security helps foster peer relationships and mitigate bullying [31]. In China, where academic pressures are high, teachers also provide a sense of stability amidst the emotional challenges adolescents face due to academic and social expectations.

The quality of teacher-student relationships has been shown to correlate with levels of peer aggression. Positive teacher-student relationships are often associated with lower levels of aggression, while negative relationships may exacerbate bullying behaviors [32]. Teacher support is a strong predictor of whether a student is likely to be bullied. For example, perceived teacher care and support can significantly reduce the likelihood of bullying victimization [33]. However, it is still unclear whether these effects persist over time, particularly in culturally distinct educational settings like China, where teacher authority and academic pressures may influence the dynamics of teacher-student relationships.

Previous research on the longitudinal impact of teacher-student relationships has yielded mixed findings. A cross-lagged study found that supportive teacher-student relationships predicted reduced victimization over time [32]. In contrast, other studies found no significant longitudinal impact, suggesting that teacher-student relationships might not consistently influence peer aggression [34]. Furthermore, teacher favoritism can lead to jealousy among peers, paradoxically increasing bullying victimization despite positive teacher-student relationships [35]. In China, where teacher authority is particularly strong, favoritism can create social divisions among students, exacerbating the risk of bullying. These conflicting findings suggest that the influence of teacher-student relationships on bullying victimization is complex and warrants further exploration across different cultural and developmental contexts. In the Chinese educational environment, where academic performance plays a significant role in social dynamics, teacher-student relationships may have unique implications for bullying victimization.

Many studies have utilized cross-lagged models to explore the relationship between teacher-student relationships and bullying [32-34]. While these models provide insights into the direction of causality, they do not adequately address the specific roles of initial levels and growth rates of key variables. This gap in the literature highlights the need for a more nuanced understanding of how teacher-student relationships evolve over time and their differential effects on bullying. Specifically, initial levels of teacher support and the growth trajectory of these relationships may exert distinct influences, potentially explaining inconsistencies in prior findings.

Therefore, this study aims to explore the impact of teacher-student relationships on bullying victimization by analyzing both initial levels and growth rates using latent growth models. This approach will provide a deeper understanding of how the dynamics of teacher-student relationships contribute to bullying prevention and how these influences evolve over time, particularly in the context of China’s academic pressures and teacher authority.

1.4. The Current Research

In summary, our study fills a gap in the literature by examining the longitudinal change trajectories of perceived control, teacher-student relationships, and bullying victimization among Chinese early adolescents. It specifically aims to investigate how teacher-student relationships mediate the relationship between perceived control and bullying victimization over time. The hypotheses are as follows: (1) Bullying victimization in early adolescents will decrease over time, while perceived control and teacher-student relationships will increase. (2) Changes in perceived control will directly predict changes in bullying victimization, and this relationship will be mediated by teacher-student relationships. (3) The initial level of teacher-student relationships positively predicts adolescents' experiences of bullying victimization. (4) Over the long term, teacher-student relationships will negatively predict bullying victimization in early adolescents.

  1. Discussion

This study, conducted over year with three waves of longitudinal tracking, found that perceived control and teacher-student relationships showed a linear increase over time, while bullying victimization decreased. The initial level of perceived control significantly predicted the initial level of bullying victimization and indirectly influenced it through the initial level of teacher-student relationships. The increasing trend in perceived control also indirectly contributed to the decreasing trend in bullying victimization, but only through the improving teacher-student relationships. This suggests that adolescents with initially higher perceived control are less likely to experience bullying victimization. However, better teacher-student relationships at the outset were associated with a higher risk of bullying. Over time, adolescents' perceived control strengthens, leading to improved teacher-student relationships, which in turn further reduced the likelihood of bullying victimization.

4.1. The Developmental Trajectories of Early Adolescent Perceived Control, Teacher-Student Relationships, and Bullying Victimization

This study found that perceived control exhibited a linear increasing trend over time, suggesting that adolescents' perceived control strengthens as they progress through early adolescence. This finding can be understood from a developmental perspective, considering the significant cognitive, emotional, and social changes that occur during this stage of life [2]. As adolescents' self-awareness matures, they develop a better understanding of their surroundings and the factors influencing their lives, which in turn enhances their perception of control [42]. Cognitive advances, such as improved executive functioning and decision-making abilities, enable adolescents to plan, strategize, and evaluate their actions more effectively, which reinforces their sense of control [43]. Social influences also contribute to the increase in perceived control. During adolescence, societal expectations encourage young people to take greater responsibility for their own actions. As adolescents face increased pressure to become more independent, they are motivated to adopt proactive behaviors, which further enhances their perceived control [44]. Moreover, positive school experiences and supportive peer interactions contribute to adolescents' confidence in their ability to influence outcomes in both academic and social contexts. Together, these factors support the growth in perceived control during adolescence.

Teacher-student relationships also showed a linear increasing trend over time, suggesting that these relationships tend to become more supportive and constructive as adolescents mature. This finding is consistent with prior research, which highlights the evolving nature of teacher-student interactions, especially as students transition into middle school [45,46]. During this period, adolescents face increased academic challenges, prompting them to rely more on teachers for both academic and emotional support. Teachers often take on quasi-parental roles, providing guidance and emotional reassurance, which helps adolescents navigate the complexities of this developmental stage [30]. As adolescents' social and emotional skills improve, they tend to place more value on these teacher-student relationships, recognizing teachers as critical sources of support. This shift reflects adolescents' growing understanding of authority figures as sources of guidance, rather than merely enforcers of rules, which may foster academic engagement and emotional resilience [47].

Lastly, bullying victimization among early adolescents demonstrated a linear decreasing trend over time, in contrast to the increases observed in perceived control and teacher-student relationships. This decline in victimization can be attributed to several developmental factors. As adolescents mature, cognitive changes such as improved empathy and perspective-taking reduce the likelihood of bullying. Adolescents' growing ability to understand the emotions and perspectives of others decreases their tendency to engage in aggressive behaviors [48]. Additionally, the development of social problem-solving skills, often promoted through peer interactions and social learning, provides adolescents with more effective ways to resolve conflicts without resorting to aggression [49]. As peer relationships become more central, adolescents prioritize maintaining harmony, which discourages aggressive behaviors. Positive peer interactions encourage pro-social behaviors, which help create a supportive social environment that reduces aggressive behaviors [50].

4.2. The Longitudinal Impact of Perceived Control on Bullying Victimization

Employing latent growth models, it was observed that, in early adolescence, the initial level of perceived control negatively predicted the initial level of bullying victimization, aligning with previous research findings [16,50-53]. Early adolescents who perceive a lack of control over their lives and studies often exhibit increased feelings of helplessness, hopelessness, and depression [54]. These emotional states can render them more vulnerable to victimization, as they may struggle to effectively manage stressful situations and defend themselves against aggression [55]. Adolescents with a stronger sense of perceived control typically exhibit higher self-esteem, confidence, and coping strategies, enabling them to confront challenges more effectively, thereby reducing their susceptibility to bullying victimization [56,57]. However, the study also found that an increasing trend in perceived control did not directly predict a decreasing trend in bullying victimization, which suggests that the relationship between these variables may not be as straightforward as previously assumed. This discrepancy could be attributed to the complex interaction between individual growth and environmental factors. Specifically, in the collectivistic context of Chinese society, adolescents may prioritize social harmony and group conformity over assertive behaviors. Despite an increase in perceived control, adolescents may be less inclined to confront bullying directly or assertively due to cultural norms that emphasize fitting in, avoiding conflict, and maintaining social cohesion within peer groups [58]. Thus, while increasing perceived control may enhance adolescents' ability to cope with stress and challenges, it may not automatically translate into a reduction in bullying victimization in contexts where social harmony and conformity are prioritized over individual assertion.

4.3. The Longitudinal Mediating Role of Teacher-Student Relationships Between Perceived Control and Bullying Victimization

This study found that the initial level of perceived control positively predicted the initial level of teacher-student relationships, and the developmental trend of perceived control positively predicted the developmental trend of teacher-student relationships, consistent with previous research [59]. Adolescents with lower perceived control may have experienced heightened negative affect, including feelings of anger and anxiety, which can accumulate over time and manifest as frustration or maladaptive behaviors [60]. These behaviors may, in turn, erode the quality of their interactions with teachers, positioning teacher-student relationships as a mediating variable influenced by changes in perceived control and potentially shaping subsequent behavioral outcomes [61].

Interestingly, this study found that the initial quality of teacher-student relationships positively predicted the initial level of bullying victimization. This finding suggests that students with stronger initial relationships with teachers may experience increased peer victimization, potentially due to social dynamics such as peer jealousy or exclusion. This finding aligns with the observations by Bokkel et al. [35] and Demol et al. [32], who noted that students perceived as having preferential relationships with teachers can become targets of peer hostility in competitive or high-pressure educational settings. Within such contexts, forming strong bonds with teachers may inadvertently distinguish these students from their peers, leading to social tension and increased bullying risk. Thus, while supportive teacher-student relationships offer critical emotional and academic benefits, they may also generate unintended social consequences in peer interactions. In terms of rate of change, an increase in teacher-student relationships was associated with a decline in bullying victimization rates. Adolescents with high-quality teacher-student relationships are more likely to develop adaptive conflict resolution skills and benefit from consistent emotional support, thereby mitigating experiences of victimization [62]. Close relationships with teachers can foster a sense of safety and trust, enhancing students' willingness to disclose bullying experiences and seek assistance [31,63]. This, in turn, allows teachers to recognize and address victimization more effectively, providing timely intervention and emotional support [62,64].

The findings of the longitudinal mediation model indicate that perceived control exerts an indirect influence on bullying victimization through its impact on teacher-student relationships. Adolescents with a stronger sense of perceived control are more likely to establish positive and supportive relationships with their teachers, which serve as protective factors against bullying. This underscores the importance of interventions aimed at fostering adolescents' perceived control, as such efforts can strengthen teacher-student bonds and reduce the likelihood of victimization. Adolescents with high perceived control are generally better equipped to manage stressors, engage in positive social interactions, and develop resilience, thereby decreasing their vulnerability to bullying [65]. Strengthening teacher-student relationships thus emerges as a critical component in helping adolescents navigate stress, enhance their coping capacity, and ultimately reduce bullying victimization.

These additions provide a comprehensive cultural context to the study, aligning the findings with the unique aspects of the Chinese educational system and its influence on adolescent experiences with bullying. Thank you again for the valuable feedback.

Comments 2: Why are are there are no control variables?

Response 2: 

Agree. We have revised the manuscript to emphasize the differences between men and women in the context of bullying victimization, perceived control, teacher-student relationships, and the mediation model.

Specifically:

Bullying Victimization and Perceived Control: After incorporating gender as a control variable, the model fit for bullying victimization and perceived control remained satisfactory, with the following fit indices:

Bullying victimization: χ²/df = 2.16, p = 0.12, CFI = 0.97, TLI = 0.92, RMSEA = 0.03, SRMR = 0.08. Perceived control: χ²/df = 1.97, p = 0.14, CFI = 0.98, TLI = 0.93, RMSEA = 0.03, SRMR = 0.05 These results suggest that gender does not significantly deteriorate the model fit for bullying victimization or perceived control.

Teacher-Student Relationships and Mediation Model: In contrast, after including gender as a control variable, the model fit for teacher-student relationships (χ²/df = 22.61, p < 0.05, CFI = 0.90, TLI = 0.71, RMSEA = 0.13, SRMR = 0.06) and the mediation model (χ²/df = 18.68, p < 0.05, CFI = 0.88, TLI = 0.79, RMSEA = 0.12, SRMR = 0.07) significantly declined. Given this significant decline in model fit, we refrain from further interpretation of these models.

The changes can be found in the revised manuscript:

Page 6, lines 270-276: "After incorporating the control variables, the model fit remains satisfactory, with χ²/df=2.16, p=0.12, CFI=0.97, TLI=0.92, RMSEA=0.03, SRMR=0.08. Gender had no statistically significant effect on the intercept of bullying victimization (β=0.014, p=0.76). Gender had a significant positive effect on the slope of bullying victimization (β=0.130, p<0.001), suggesting that females experience a higher rate of decrease in bullying victimization compared to males."

Page 7, lines 286-291: "After incorporating the control variables, the model fit remains satisfactory, with χ²/df=1.97, p=0.14, CFI=0.98, TLI=0.93, RMSEA=0.03, SRMR=0.05. Gender had a significant negative effect on the intercept of perceived control (β=-0.24, p<0.001), indicating that females have a lower initial level of perceived control compared to males. Gender had a significant positive effect on the slope of perceived control (β=0.12, p<0.001), suggesting that females have a higher rate of increase in perceived control compared to males."

Page 7, lines 301-302: "After incorporating the control variables, the model fit significantly declined, with χ²/df=22.61, p<0.05, CFI=0.90, TLI=0.71, RMSEA=0.13, SRMR=0.06."

Page 7, lines 310-311: "After incorporating the control variables, the model fit significantly declined, with χ²/df=18.68, p<0.05, CFI=0.88, TLI=0.79, RMSEA=0.12, SRMR=0.07."

We hope this addresses your concern. Thank you for your valuable suggestion.

Comments 3: Response to Comments on the Quality of English Language

Thank you for this important feedback. We appreciate your thorough review and constructive comments. We acknowledge that clarity and conciseness are crucial for effectively communicating our findings. Therefore, we have revised the manuscript to address issues of readability, duplication, and redundancy, ensuring a more streamlined presentation of key concepts and findings.

We hope that these revisions address your concerns and enhance the clarity of our study. The revised manuscript is attached for your review.

Once again, thank you for your valuable feedback. We hope our revisions adequately address the points you raised. Wishing you continued success in your academic endeavors and a pleasant day ahead.

Sincerely

Reviewer 2 Report

Comments and Suggestions for Authors

Thank you for the opportunity to review the paper.

It might be advisable to explain how the tests were conducted at different stages.

It is important to provide data on whether there are differences between men and women.

Author Response

Dear Reviewer,

Thank you for your thoughtful and constructive feedback on our manuscript. We appreciate the time and effort you have taken to provide valuable suggestions that have helped us improve the clarity and rigor of our study. In response to your comments, we have made the following revisions:

Comments 1: It might be advisable to explain how the tests were conducted at different stages.

Response 1:

Thank you for pointing this out. I agree with this comment. Therefore, I have expanded the description to clarify the process and stages of the study. Specifically, I have provided additional information about the guidance provided by two trained graduate students in psychology during the testing, and the option for participants to withdraw at any time. The revised details also emphasize that the questionnaires were completed in classrooms during regular class sessions, within a 30-minute time frame.

The changes can be found in the revised manuscript on page 5, lines 199-206.

Updated text in the manuscript:

This study was approved by the Ethics Committee of the first author’s University (Ethical Approval No. ZZUIRB2020-021). Consent was obtained from both parents and students prior to the survey. At each wave, questionnaires were administered in class-rooms during regular class sessions, with students completing them independently under the guidance of two trained graduate students in psychology. The students were supervised during the process, which lasted no more than 30 minutes. Participants were informed that they could withdraw from the study at any time without penalty. All collected data were directly encrypted and stored by the first author.

Comments 2: It is important to provide data on whether there are differences between men and women.

Agree. We have revised the manuscript to emphasize the differences between men and women in the context of bullying victimization, perceived control, teacher-student relationships, and the mediation model. Specifically, The model fit for bullying victimization and perceived control remained satisfactory, even with the inclusion of gender (χ²/df = 2.16, p = 0.12, CFI = 0.97, TLI = 0.92, RMSEA = 0.03, SRMR = 0.08 for bullying victimization; χ²/df = 1.97, p = 0.14, CFI = 0.98, TLI = 0.93, RMSEA = 0.03, SRMR = 0.05 for perceived control). These results suggest that gender does not significantly deteriorate the model fit for bullying victimization or perceived control. In contrast, after including gender as a control variable, the model fit for teacher-student relationships (χ²/df = 22.61, p < 0.05, CFI = 0.90, TLI = 0.71, RMSEA = 0.13, SRMR = 0.06) and the mediation model (χ²/df = 18.68, p < 0.05, CFI = 0.88, TLI = 0.79, RMSEA = 0.12, SRMR = 0.07) significantly declined. Given this significant decline in model fit, we refrain from further interpretation of these models.

The changes can be found in the revised manuscript on page 6, lines 270-276.

Updated text in the manuscript:

After incorporating the control variables, the model fit remains satisfactory, with χ²/df=2.16, p=0.12, CFI=0.97, TLI=0.92, RMSEA=0.03, SRMR=0.08. Gender had no statistically significant effect on the intercept of bullying victimization (β=0.014, p=0.76). Gender had a significant positive effect on the slope of bullying victimization (β=0.130, p<0.001), suggesting that females experience a higher rate of decrease in bullying victimization compared to males.

The changes can be found in the revised manuscript on page 7, lines 286-291.

Updated text in the manuscript:

After incorporating the control variables, the model fit remains satisfactory, with χ²/df=1.97, p=0.14, CFI=0.98, TLI=0.93, RMSEA=0.03, SRMR=0.05. Gender had a significant negative effect on the intercept of perceived control (β=-0.24, p<0.001), indicating that females have a lower initial level of perceived control compared to males. Gender had a significant positive effect on the slope of perceived control (β=0.12, p<0.001), suggesting that females have a higher rate of increase in perceived control compared to males.

The changes can be found in the revised manuscript on page 7, lines 301-302.

Updated text in the manuscript:

After incorporating the control variables, the model fit significantly declined, with χ²/df=22.61, p<0.05, CFI=0.90, TLI=0.71, RMSEA=0.13, SRMR=0.06.

The changes can be found in the revised manuscript on page 7, lines 310-311.

Updated text in the manuscript:

After incorporating the control variables, the model fit significantly declined, with χ²/df=18.68, p<0.05, CFI=0.88, TLI=0.79, RMSEA=0.12, SRMR=0.07.

We hope that these revisions address your concerns and enhance the clarity of our study. We truly appreciate your input and believe these changes significantly strengthen the manuscript.

The revised manuscript is attached for your review.

Once again, thank you for your valuable feedback. We hope our revisions adequately address the points you raised. Wishing you continued success in your academic endeavors and a pleasant day ahead.

Reviewer 3 Report

Comments and Suggestions for Authors

Dear authors,

Congratuation for explore an importnt issue. 

I suggest the authors to rearenge the discussion improving with referecens from introduction. 

Please see my comments at the attached file. 

Author Response

Dear Reviewer,

Thank you for your positive feedback and valuable suggestions. We appreciate your acknowledgment of the importance of the issue explored in our study. Your recommendation to improve the discussion by referencing key points from the introduction is highly appreciated, and we agree that this would enhance the coherence of the manuscript.

In response to your comments, we have made the following revisions:

Comments 1: I suggest the authors to rearenge the discussion improving with referecens from introduction.

Thank you for pointing this out. We agree with this comment. Therefore, we have strengthened the connection between the discussion and the introduction by incorporating references and key theoretical frameworks discussed earlier in the paper. Specifically:

4.1. The Developmental Trajectories of Early Adolescent Perceived Control, Teacher-Student Relationships, and Bullying Victimization

This study found that perceived control exhibited a linear increasing trend over time, suggesting that adolescents' perceived control strengthens as they progress through early adolescence. This finding can be understood from a developmental perspective, considering the significant cognitive, emotional, and social changes that occur during this stage of life [2]. As adolescents' self-awareness matures, they develop a better understanding of their surroundings and the factors influencing their lives, which in turn enhances their perception of control [42]. Cognitive advances, such as improved executive functioning and decision-making abilities, enable adolescents to plan, strategize, and evaluate their actions more effectively, which reinforces their sense of control [43]. Social influences also contribute to the increase in perceived control. During adolescence, societal expectations encourage young people to take greater responsibility for their own actions. As adolescents face increased pressure to become more independent, they are motivated to adopt proactive behaviors, which further enhances their perceived control [44]. Moreover, positive school experiences and supportive peer interactions contribute to adolescents' confidence in their ability to influence outcomes in both academic and social contexts. Together, these factors support the growth in perceived control during adolescence.

Teacher-student relationships also showed a linear increasing trend over time, suggesting that these relationships tend to become more supportive and constructive as adolescents mature. This finding is consistent with prior research, which highlights the evolving nature of teacher-student interactions, especially as students transition into middle school [45,46]. During this period, adolescents face increased academic challenges, prompting them to rely more on teachers for both academic and emotional support. Teachers often take on quasi-parental roles, providing guidance and emotional reassurance, which helps adolescents navigate the complexities of this developmental stage [30]. As adolescents' social and emotional skills improve, they tend to place more value on these teacher-student relationships, recognizing teachers as critical sources of support. This shift reflects adolescents' growing understanding of authority figures as sources of guidance, rather than merely enforcers of rules, which may foster academic engagement and emotional resilience [47].

Lastly, bullying victimization among early adolescents demonstrated a linear decreasing trend over time, in contrast to the increases observed in perceived control and teacher-student relationships. This decline in victimization can be attributed to several developmental factors. As adolescents mature, cognitive changes such as improved empathy and perspective-taking reduce the likelihood of bullying. Adolescents' growing ability to understand the emotions and perspectives of others decreases their tendency to engage in aggressive behaviors [48]. Additionally, the development of social problem-solving skills, often promoted through peer interactions and social learning, provides adolescents with more effective ways to resolve conflicts without resorting to aggression [49]. As peer relationships become more central, adolescents prioritize maintaining harmony, which discourages aggressive behaviors. Positive peer interactions encourage pro-social behaviors, which help create a supportive social environment that reduces aggressive behaviors [50].

4.2. The Longitudinal Impact of Perceived Control on Bullying Victimization

Employing latent growth models, it was observed that, in early adolescence, the initial level of perceived control negatively predicted the initial level of bullying victimization, aligning with previous research findings [16,50-53]. Early adolescents who perceive a lack of control over their lives and studies often exhibit increased feelings of helplessness, hopelessness, and depression [54]. These emotional states can render them more vulnerable to victimization, as they may struggle to effectively manage stressful situations and defend themselves against aggression [55]. Adolescents with a stronger sense of perceived control typically exhibit higher self-esteem, confidence, and coping strategies, enabling them to confront challenges more effectively, thereby reducing their susceptibility to bullying victimization [56,57]. However, the study also found that an increasing trend in perceived control did not directly predict a decreasing trend in bullying victimization, which suggests that the relationship between these variables may not be as straightforward as previously assumed. This discrepancy could be attributed to the complex interaction between individual growth and environmental factors. Specifically, in the collectivistic context of Chinese society, adolescents may prioritize social harmony and group conformity over assertive behaviors. Despite an increase in perceived control, adolescents may be less inclined to confront bullying directly or assertively due to cultural norms that emphasize fitting in, avoiding conflict, and maintaining social cohesion within peer groups [58]. Thus, while increasing perceived control may enhance adolescents' ability to cope with stress and challenges, it may not automatically translate into a reduction in bullying victimization in contexts where social harmony and conformity are prioritized over individual assertion.

4.3. The Longitudinal Mediating Role of Teacher-Student Relationships Between Perceived Control and Bullying Victimization

This study found that the initial level of perceived control positively predicted the initial level of teacher-student relationships, and the developmental trend of perceived control positively predicted the developmental trend of teacher-student relationships, consistent with previous research [59]. Adolescents with lower perceived control may have experienced heightened negative affect, including feelings of anger and anxiety, which can accumulate over time and manifest as frustration or maladaptive behaviors [60]. These behaviors may, in turn, erode the quality of their interactions with teachers, positioning teacher-student relationships as a mediating variable influenced by changes in perceived control and potentially shaping subsequent behavioral outcomes [61].

Interestingly, this study found that the initial quality of teacher-student relationships positively predicted the initial level of bullying victimization. This finding suggests that students with stronger initial relationships with teachers may experience increased peer victimization, potentially due to social dynamics such as peer jealousy or exclusion. This finding aligns with the observations by Bokkel et al. [35] and Demol et al. [32], who noted that students perceived as having preferential relationships with teachers can become targets of peer hostility in competitive or high-pressure educational settings. Within such contexts, forming strong bonds with teachers may inadvertently distinguish these students from their peers, leading to social tension and increased bullying risk. Thus, while supportive teacher-student relationships offer critical emotional and academic benefits, they may also generate unintended social consequences in peer interactions. In terms of rate of change, an increase in teacher-student relationships was associated with a decline in bullying victimization rates. Adolescents with high-quality teacher-student relationships are more likely to develop adaptive conflict resolution skills and benefit from consistent emotional support, thereby mitigating experiences of victimization [62]. Close relationships with teachers can foster a sense of safety and trust, enhancing students' willingness to disclose bullying experiences and seek assistance [31,63]. This, in turn, allows teachers to recognize and address victimization more effectively, providing timely intervention and emotional support [62,64].

The findings of the longitudinal mediation model indicate that perceived control exerts an indirect influence on bullying victimization through its impact on teacher-student relationships. Adolescents with a stronger sense of perceived control are more likely to establish positive and supportive relationships with their teachers, which serve as protective factors against bullying. This underscores the importance of interventions aimed at fostering adolescents' perceived control, as such efforts can strengthen teacher-student bonds and reduce the likelihood of victimization. Adolescents with high perceived control are generally better equipped to manage stressors, engage in positive social interactions, and develop resilience, thereby decreasing their vulnerability to bullying [65]. Strengthening teacher-student relationships thus emerges as a critical component in helping adolescents navigate stress, enhance their coping capacity, and ultimately reduce bullying victimization.

These changes ensure a more coherent connection between the discussion and the introduction, as suggested. Thank you again for your valuable feedback.

Comments 2: Please see my comments at the attached file.

Response 2: 

Based on your comment to "please add some references" .

The changes made are as follows:

Many studies have utilized cross-lagged models to explore the relationship between teacher-student relationships and bullying [32-34].

This revision can be found in the revised manuscript on Page 4, Line 176-177.

Based on your comment to "please add information regarding age" .

The changes made are as follows:

A total of 1246 participants (Mage = 13.58 years, SD = 0.75) who completed all three assessments were retained, including 601 males (48.23%) and 645 females (51.77%).

This revision can be found in the revised manuscript on Page 4, Line 194-196.

Based on your comment to "please add an explication because the number decreased" .

The changes made are as follows:

A total of 208 students were unable to complete the full assessment due to illness, temporary withdrawal, transfer, or other related reasons.

This revision can be found in the revised manuscript on Page 4, Line 191-193.

Based on your comment to "identify the instructions that were given regarding which teacher the student had to report to" we have clarified the instructions in the manuscript.

The changes made are as follows:

The Teacher-Student Relationship Scale(TSRS) was employed for assessment, with students reporting their responses through self-assessment.

This revision can be found in the revised manuscript on Page 5, Line 219-220.

Based on your comment to "did the authors add all itens to have a total score? Conflict it is a negative dimention." .

The changes made are as follows:

This scale includes dimensions of closeness, conflict (reverse-scored), support, and satisfaction, comprising 23 items scored on a 5-point Likert scale ranging from 1 (totally disagree) to 5 (totally agree).

This revision can be found in the revised manuscript on Page 5, Line 222-224.

Based on your recommendation to "rearrange the sentence in order to clarify (changes three times)," we have revised the sentence to reduce the repetition of the word "changes" and to make the meaning clearer.

The changes made are as follows:

We restructured the sentence to avoid redundancy and improve clarity. Specifically, we reworded the sentence to indicate that changes in perceived control influence bullying through changes in teacher-student relationships.

This revision can be found in the revised manuscript on Page 7, Line 302-303.

Based on your comment to "please add practical implications of the study" .

The changes made are as follows:

This study contributes to understanding bullying victimization among adolescents by identifying patterns of change and highlighting key risk factors. The findings offer practical implications for educational practice and policy. First, the positive association between perceived control and reduced bullying victimization underscores the importance of school programs that enhance students' sense of agency and self-efficacy, such as social-emotional learning (SEL) programs aimed at building resilience and coping skills. Second, the mediating role of teacher-student relationships suggests that fostering positive interactions between teachers and students is crucial. Training teachers to cultivate empathetic, supportive connections with students can create a protective school climate and encourage students to seek help when facing bullying. Additionally, recognizing that strong teacher-student bonds may lead to social tension among peers highlights the need for strategies that build inclusive peer environments, promoting cohesion and reducing jealousy-driven dynamics.

This revision can be found in the revised manuscript on Page 11, Line 473-485.

We hope that these revisions address your concerns and enhance the clarity of our study. The revised manuscript is attached for your review.

Once again, thank you for your valuable feedback. We hope our revisions adequately address the points you raised. Wishing you continued success in your academic endeavors and a pleasant day ahead.

Sincerely

Round 2

Reviewer 1 Report

Comments and Suggestions for Authors

·         Abstract: “existing research” is twice in line 2

·         Abstract: “This study endeavors…” It sounds like you are exploring how bullying can be protective, which is odd.

·         Abstract overall needs to be reworked. It is repetitive with language.

·         Line 39: Is this 30.4% worldwide, the US, China, or somewhere else?

·         Great context addition in the intro!

·         Line 51: Predictive or protective?

·         Line 157: Exacerbate engaging in bullying or being a victim of bullying?

·         What are the average ages of the sample in each wave? This seems very important.

·         Line 317: Slower rate of improvement in what?

·         Line 318: Model fit significantly declined – What does this mean for gender?

·         Line 327: Model fit significantly declined – What does this mean for gender?

·         Line 351: Year or years? How many years?

·         Lines 360-362: This is a little confusing. If the relationships are good at the beginning, but that causes bullying, then why does bullying decrease when the relationships get even better? – In the section starting on line 430, this contradiction does not seem properly noted or addressed.

·         Policy section needs citations.

Author Response

Dear Reviewer,

We would like to sincerely thank you for your detailed and thoughtful review of our manuscript. Your insightful comments and constructive suggestions have been incredibly valuable in improving the quality of our work. We deeply appreciate the time and effort you took to provide such comprehensive feedback.

We have carefully addressed each of your suggestions and made the necessary revisions to enhance the clarity and depth of our study. Your feedback has significantly contributed to refining our paper, and we believe it is now much stronger as a result.

In response to your comments, we have made the following revisions:

Comments 1: Abstract: “existing research” is twice in line 2

Response 1: Thank you for pointing out the repetition of "existing research" in line 2 of the abstract. We have revised the sentence to eliminate the redundancy. The updated version now reads as follows (lines 9-11):

“However, existing research often lacks a longitudinal perspective and sufficient attention to protective factors, particularly the dynamic role of teacher-student relationships.”

This revision addresses your concern and improves the clarity of the abstract.

Comments 2: Abstract: “This study endeavors…” It sounds like you are exploring how bullying can be protective, which is odd.

Response 2: Thank you for your valuable feedback. We understand that the phrase “This study endeavors…” might suggest an exploration of how bullying could be protective, which was not the intended focus. To clarify, the study aims to examine the protective factors against bullying victimization, specifically the roles of perceived control and teacher-student relationships.

We have revised this sentence for better clarity (lines 11-13):

“This study explores the longitudinal protective mechanisms against bullying victimization, focusing on the roles of perceived control and teacher-student relationships.”

This revision should more accurately convey the objective of our study.

Comments 3: Abstract overall needs to be reworked. It is repetitive with language.

Response 3: Thank you for your constructive feedback regarding the overall repetition in the abstract. We have carefully reviewed the text and made revisions to reduce redundancy while maintaining clarity and precision. Specifically, we have:

Removed repeated phrases and restructured sentences to eliminate unnecessary repetition.

Improved the flow of the abstract by simplifying certain sections and making the key points more concise.

The revised abstract now presents the essential information in a more streamlined and focused manner. We believe these changes address your concern and improve the overall quality of the abstract.

The changes can be found in the revised manuscript:

Page 1, lines 9-26:

Abstract: Background: Bullying victimization remains a grave issue in early adolescence. However, existing research often lacks a longitudinal perspective and sufficient attention to protective factors, particularly the dynamic role of teacher-student relationships. Objective: This study explores the longitudinal protective mechanisms against bullying victimization, focusing on the roles of per-ceived control and teacher-student relationships. Methods: A sample of 1,454 adolescents (Mean age = 13.63 years, SD = 0.76, 51% female) was followed over the course of one year. Latent growth models were employed to examine the developmental trajectories of bullying victimization, per-ceived control, and teacher-student relationships, and to investigate the longitudinal mediating role of teacher-student relationships in the relationship between perceived control and bullying victimization. Results: Both perceived control and teacher-student relationships showed a con-sistent increase over time, while bullying victimization declined over time. The intercept of per-ceived control negatively predicted the intercept of bullying victimization, with this relationship mediated by the intercept of teacher-student relationships. Furthermore, the slope of perceived control affected the slope of bullying victimization solely through the slope of teacher-student relationships. Conclusions: These findings highlight that perceived control serves as a stable pro-tective factor against bullying, while strong teacher-student relationships can further reduce bul-lying victimization. Enhancing students' perceived control and fostering supportive teach-er-student relationships should be key components of anti-bullying interventions.

Comments 4: Line 39: Is this 30.4% worldwide, the US, China, or somewhere else?

Response 4: Thank you for your helpful comment. To clarify, the 30.4% figure refers to the global prevalence of bullying behaviors. We have revised the sentence to specify this point by adding "worldwide" to ensure the scope is clear.

The updated sentence now reads (lines 38-40):

"This developmental stage also coincides with a peak in the prevalence of bullying behaviors, with approximately 30.4% of early adolescents worldwide experiencing bullying."

We hope this revision addresses your concern.

Comments 5: Line 51: Predictive or protective?

Response 5: Thank you for pointing out the terminology. We agree that "protective" is a more appropriate term than "predictive" in this context. We have revised the sentence accordingly (lines 50-51):

"Existing research indicates that perceived control and teacher-student relationships may serve as effective protective factors against bullying victimization."

This change better reflects the intended meaning of the study.

Comments 6: Line 157: Exacerbate engaging in bullying or being a victim of bullying?

Response 6: Thank you for your insightful suggestion. We now understand that the focus should be on how negative teacher-student relationships may exacerbate being a victim of bullying, rather than bullying behaviors. We have revised the sentence accordingly (lines 156-158):

"Positive teacher-student relationships are often associated with lower levels of aggression, while negative relationships may exacerbate being a victim of bullying."

This revision better captures the intended relationship between negative teacher-student interactions and victimization in bullying.

Comments 7: What are the average ages of the sample in each wave? This seems very important.

Response7: Thank you for your valuable comment. We agree that the average ages of the sample in each wave are important details to include. We have added this information to the manuscript to provide more clarity on the sample characteristics. The revised description now includes the average ages for each wave as follows (lines 207-209):

"The number of participants was 1,414 (Mage = 13.63 years, SD = 0.76, June 2022), 1,328 (Mage = 14.04 years, SD = 0.73, December 2022), and 1,246 (Mage = 14.66 years, SD = 1.14, June 2023), respectively, for each wave."

We hope this revision addresses your concern.

Comments 8: Line 317: Slower rate of improvement in what?

Response 8 : Thank you for your helpful comment. We agree that the sentence should be more specific about the "rate of improvement." We have revised it to clarify that the slower rate of improvement refers to the improvement in teacher-student relationships (lines 317-319):

"The correlation between the intercept and slope was significant (r = -0.31, p<0.01), suggesting that a higher initial level of teacher-student relationships was associated with a slower rate of improvement in teacher-student relationships."

This revision should provide clearer meaning and improve the precision of the statement.

Comments 9: Line 318: Model fit significantly declined – What does this mean for gender?

Response 9: Thank you for your comment. We have revised the manuscript to clarify that the decline in model fit may be due to the weak correlation between gender and the dependent variable. Including gender as a control variable likely introduced noise, reducing the model’s explanatory power.

The revised sentence now reads (lines 321-323):

This decline suggests that including gender as a control variable, due to its weak correlation with the dependent variable, added noise to the model and reduced its explanatory power.

We hope this revision addresses your concern.

Comments 10: Line 327: Model fit significantly declined – What does this mean for gender?

Response 10: Thank you for your comment. We have revised the manuscript to clarify that the decline in model fit may be due to the weak correlation between gender and the dependent variable. Including gender as a control variable likely added minimal value and introduced unnecessary complexity, which negatively affected the model fit.

The revised sentence now reads (line 333-335):

We hope this revision better addresses your concern.

Comments 11: Line 351: Year or years? How many years?

Response 11: Thank you for your comment. We have revised the sentence to clarify that the study was conducted over one year (lines 359-361):

This study, conducted over one year with three waves of longitudinal tracking, found that perceived control and teacher-student relationships showed a linear increase over time, while bullying victimization decreased.

Comments 12: Lines 360-362: This is a little confusing. If the relationships are good at the beginning, but that causes bullying, then why does bullying decrease when the relationships get even better? – In the section starting on line 430, this contradiction does not seem properly noted or addressed.

Response 12: Thank you for your thoughtful comment.

We have revised the manuscript to clarify the relationship between teacher-student relationships and bullying victimization (lines 449-467):

Interestingly, this study found that the initial quality of teacher-student relation-ships positively predicted the initial level of bullying victimization. This finding suggests that students with stronger initial relationships with teachers may experience increased peer victimization, potentially due to social dynamics such as peer jealousy or exclusion. This finding aligns with the observations by Bokkel et al. [35] and Demol et al. [32], who noted that students perceived as having preferential relationships with teachers can become targets of peer hostility in competitive or high-pressure educational settings. Within such contexts, forming strong bonds with teachers may inadvertently distinguish these students from their peers, leading to social tension and increased bullying risk. Thus, while supportive teacher-student relationships offer critical emotional and aca-demic benefits, they may also generate unintended social consequences in peer interac-tions. In terms of rate of change, an increase in teacher-student relationships was asso-ciated with a decline in bullying victimization rates. Adolescents with high-quality teacher-student relationships are more likely to develop adaptive conflict resolution skills and benefit from consistent emotional support, thereby mitigating experiences of victimization [62]. Close relationships with teachers can foster a sense of safety and trust, enhancing students' willingness to disclose bullying experiences and seek assistance [31,63]. This, in turn, allows teachers to recognize and address victimization more effec-tively, providing timely intervention and emotional support [62,64].

We hope this revision addresses the concern and clarifies the complex dynamics between teacher-student relationships and bullying.

Comments 13: Policy section needs citations.

Response 13 : Thank you for your valuable feedback. We have now revised the policy section by adding appropriate citations to support the recommendations made.

  1. Yang, C.; Chan, M.K.; Ma, T.L. School-wide social emotional learning (SEL) and bullying victimization: Moderating role of school climate in elementary, middle, and high schools. J Sch Psychol 2020, 82, 49-69. https://doi.org/10.1016/j.jsp.2020.08.002.
  2. Van Ryzin, M.J.; Roseth, C.J. Cooperative Learning in Middle School: A Means to Improve Peer Relations and Reduce Vic-timization, Bullying, and Related Outcomes. J Educ Psychol 2018, 110, 1192-1201. https://doi.org/10.1037/edu0000265.

We believe this addition strengthens the foundation of the policy suggestions and ensures they are grounded in existing research and best practices.

Once again, thank you for your hard work and dedication. I wish you all the best in both your personal and academic endeavors.

Warm regards,
